# Data Pruning via Moving-one-Sample-out

**Haoru Tan**[1, 3] *    **Sitong Wu**[2, 3] *    **Fei Du**[3, 4]

**Yukang Chen**[2]    **Zhibin Wang**[3, 4]    **Fan Wang**[3, 4]    **Xiaojuan Qi**[1]

[1]HKU    [2]CUHK    [3]DAMO Academy, Alibaba Group    [4]Hupan Lab, Zhejiang Province

Official Implementation

## Abstract

In this paper, we propose a novel data-pruning approach called moving-one-sample-out (MoSo), which aims to identify and remove the least informative samples from the training set. The core insight behind MoSo is to determine the importance of each sample by assessing its impact on the optimal empirical risk. This is achieved by measuring the extent to which the empirical risk changes when a particular sample is excluded from the training set. Instead of using the computationally expensive leaving-one-out-retraining procedure, we propose an efficient first-order approximator that only requires gradient information from different training stages. The key idea behind our approximation is that samples with gradients that are consistently aligned with the average gradient of the training set are more informative and should receive higher scores, which could be intuitively understood as follows: if the gradient from a specific sample is consistent with the average gradient vector, it implies that optimizing the network using the sample will yield a similar effect on all remaining samples. Experimental results demonstrate that MoSo effectively mitigates severe performance degradation at high pruning ratios and achieves satisfactory performance across various settings.

## 1 Introduction

The recent advances in AI have been largely driven by the availability of large-scale datasets [31, 3, 6, 34, 41, 10, 37], which enable the training of powerful models [51, 3, 45, 6, 1, 4, 36]. However, such datasets also pose significant challenges in terms of computational and storage resources. It is important to note that these datasets may contain redundant or noisy samples that are either irrelevant or harmful to the model's performance. Data pruning techniques aim to address these issues by removing such samples and retaining a smaller, more compact core set of training samples [11, 28, 42, 27, 48, 38, 15]. This can not only reduce the costs of model training and data storage but also maintain the performance of the model compared to the original dataset.

Existing approaches can be broadly categorized into three major groups: pruning by importance criteria [11, 42, 28, 27, 26, 46], coverage or diversity-driven methods [48, 29, 33, 15], and optimization-based methods [38, 8, 23, 24, 21, 44]. Among these, the first group of studies is the most effective and popular. These studies assume that hard samples are critical and informative core-set samples, and thus, they design difficulty-based metrics to assess sample importance. Such metrics include prediction entropy [11], forgetting [28] or memorization [46] score, gradient norm [27], E2LN (variance of prediction) [27], self-supervised prototype distance [42], diverse ensembles [26], and others.

---

*Equal contribution.

37th Conference on Neural Information Processing Systems (NeurIPS 2023).

**Limitations and Motivations.** The methods we discussed have some major drawbacks: (i). Hard samples are not necessarily important samples. For example, noisy samples [48] and outliers [39] often lead to high losses, which makes it difficult for importance criteria [11, 27] to distinguish them from truly important samples. (ii). Training dynamics are rarely considered. The mainstream methods [27, 46, 26, 11, 42, 15, 48, 38] do not possess the awareness of training dynamics as they generally utilize a converged surrogate network for data selection. This may favor samples that are difficult or influential in the later stages of training, but not necessarily in the earlier stages or the whole training process [50, 12].

**Our Method.** In this paper, we propose a new data pruning algorithm, which involves the newly proposed Moving-one-Sample-out (MoSo) score with an efficient and error-guaranteed estimator. To address the first limitation, MoSo utilizes *the change of the optimal empirical risk when removing a specific sample from the training set* to measure sample importance instead of only focusing on sample difficulty. By doing so, MoSo can better separate important samples from harmful noise samples, as the former tends to lower the empirical risk, while the latter may increase it. However, MoSo is too costly to compute exactly as it needs brute force leaving-one-out-retraining. Therefore, we propose an efficient first-order approximator with linear complexity and guaranteed approximation error. The proposed approximation is simple: samples whose gradient agrees with gradient expectations at all training stages will get higher scores, which could be intuitively understood as follows: if the gradient from a specific sample is consistent with the average gradient vector, it implies that optimizing the network using the sample will yield a similar effect on all remaining samples. The second limitation is addressed since MoSo comprehensively considers information from different training stages.

We evaluate our MoSo on CIFAR-100 [5], Tiny-ImageNet [49], and ImageNet-1K [31] under different pruning ratios. As shown in Figure 1, our MoSo significantly surpasses the previous state-of-the-art methods, especially for high pruning ratios. Besides, experimental results demonstrate that the coreset selected by our MoSo under one network (such as ResNet) can generalize well to other unseen networks (such as SENet and EfficientNet) (refer to Figure 3(a) and Figure 3(b)). Additionally, we study the robustness of our MoSo on datasets with synthetic noisy labels (refer to Figure 3(c) and Figure 3(d)). It can be seen that our MoSo performs best on average, and surpasses the previous methods based on difficulty-based importance criteria by a large margin.

## 2   Related Work

Finding important samples is not only the purpose of data pruning, but also the core step in many machine learning tasks and problems, like active learning [7, 52, 32, 39, 19], noisy learning [9], and continuous learning [16]. In data-efficient deep learning, there are also some related topics like data distillation [13, 40, 53] and data condensation [17, 47, 20]. Unlike data pruning, they focus on synthesizing a small but informative dataset as an alternative to the original large-scale dataset. Existing data selection/pruning approaches could be broadly divided into several categories, pruning by importance criteria [11, 42, 28, 27, 26, 46], coverage or diversity driven methods [48, 29, 33, 15], optimization-based methods [38, 8, 23–25, 21, 44, 43].

**Pruning by importance criteria.** This group of studies is the most popular. Generally, they assume that hard samples are critical and informative core-set samples and thus design difficulty-based metrics to assess sample importance. The EL2N score [27] measures the data difficulty by computing the average of the $\ell_2$-norm error vector from a set of networks. GraNd [27] measures the importance by calculating the expectation of the gradient norm. The Forgetting score [28] counts how many times a model changes its prediction from correct to incorrect for each example during the training process. Memorization [46] assigns a score to each example based on how much its presence or absence in the training set affects the model's ability to predict it correctly. Diverse ensembles [26] gave a score to each sample based on how many models in a group misclassified it. However, hard samples are not necessarily good for model training [42]. For example, noisy samples [48] and outliers [39] often lead to high losses, which makes it difficult for importance criteria [11, 27] to distinguish them from truly important samples. As a comparison, our MoSo score measures sample importance instead of only focusing on sample difficulty by calculating *the change of the optimal empirical risk when removing a specific sample from the training set*. By doing so, it can better separate important samples from harmful noise samples, as the former tends to lower the empirical risk, while the latter may increase it.

**Coverage or diversity driven methods.** Sener et. al. [32] applied greedy k-center to choose the coreset with good data coverage. BADGE [19] is a diversity-based selection method in active learning that clusters the gradient embeddings of the current model using k-means++ and selects a subset from each cluster. CCS [15] balances the data distribution and the example importance in selecting data points. Moderate [48] chooses data points with scores near the median score. Note that some diversity-driven methods, such as CCS [15] and Moderate [48], can use any selection criterion, such as EL2N score [27], as a basis.

**Optimization-based methods.** In addition, a line of recent works proposed to select data by optimization, such as gradient matching [8, 23], bi-level optimization [24, 25], submodularity [21, 44, 43]. One of the most advanced methods, the optimization-based dataset pruning [38], builds an algorithm over the sample-wise influence function [35] to remove samples with minimal impact and guarantees generalization. However, like mainstream methods [27, 46, 26, 11, 42, 15, 48, 38], it does not account for the effect of samples on the training dynamics, as it only uses the information from the final model. This may favor samples that are difficult or influential in the later stages of training, but not necessarily in the earlier stages or the whole training process [50, 12]. In our work, the proposed method is fully training-dynamic-aware since the MoSo's approximator comprehensively considers information from different training stages.

## 3 Method

In the following, we will first present the background knowledge in Section 3.1. Following that, we will elaborate on the proposed MoSo score for assessing sample importance in Section 3.2. Furthermore, we will introduce an efficient approximator of MoSo in Section 3.3. Section 3.4 shows the along with its complexity analysis and error guarantees.

### 3.1 Background

In this work, we focus on the classification task, where $\mathcal{S} = \{(x_i, y_i)|_{i=1}^N\}$ denotes the training set, drawn i.i.d from an underlying data distribution $P$, with input vectors $x \in \mathbb{R}^d$ and one-hot label vectors $y \in \{0, 1\}^K$. Let $l(\cdot)$ denote the widely used cross-entropy loss function for classification tasks. Given a pruning ratio $\delta$ and a parameterized deep network $f_\mathbf{w}$, the data pruning task aims to find the most representative training subset $\hat{S} \subset \mathcal{S}$ while pruning the remaining samples. This can be formulated as:

$$\hat{S} = \underset{D \subset \mathcal{S}}{\arg\min} \, \mathbb{E}_{z:(\boldsymbol{x},y) \sim P}\Big[l(z, \mathbf{w}_D^*)\Big], \tag{1}$$

where $(|\mathcal{S}| - |D|)/|\mathcal{S}| = \delta$, $|\cdot|$ represents the cardinality of a set, and $\mathbf{w}_D^*$ indicates the optimal network parameter trained on $D$ with the stochastic gradient descent (SGD) optimizer. The SGD optimizer updates the parameters as follows:

$$\mathbf{w}_t = \mathbf{w}_{t-1} - \eta_t \nabla \mathcal{L}(\mathcal{B}_t, \mathbf{w}_{t-1}), \tag{2}$$

where $t \in \{1, ..., T\}$, $\eta_t$ is the learning rate at the $t$-th step, and $\mathcal{B}_t$ represents the mini-batch, $\nabla$ is the gradient operator with respect to network parameters, $\mathcal{L}(\cdot)$ is the average cross-entropy loss on the given set/batch of samples.

### 3.2 Definition for Moving-one-Sample-out

Here, we will describe the details of Moving-one-Sample-out (MoSo) score.

**Definition 1.** *The MoSo score for a specific sample $z$ selected from the training set $\mathcal{S}$ is*

$$\mathcal{M}(z) = \mathcal{L}\Big(\mathcal{S}/z, \mathbf{w}_{\mathcal{S}/z}^*\Big) - \mathcal{L}\Big(\mathcal{S}/z, \mathbf{w}_{\mathcal{S}}^*\Big), \tag{3}$$

*where $\mathcal{S}/z$ indicates the dataset $\mathcal{S}$ excluding $z$, $\mathcal{L}(\cdot)$ is the average cross-entropy loss on the considered set of samples, $\mathbf{w}_{\mathcal{S}}^*$ is the optimal parameter trained on the full set $\mathcal{S}$, and $\mathbf{w}_{\mathcal{S}/z}^*$ is the optimal parameter on $\mathcal{S}/z$.*

The MoSo score measures the importance of a specific sample $z$ by calculating how the empirical risk over $\mathcal{S}/z$ changes when removing $z$ from the training set. Specifically, with a *representative*

*(important and with proper annotation)* sample $z$, retaining it can promote training and result in a lower empirical risk while removing it could be harmful to the training and result in a higher empirical risk. Hence, $\mathcal{M}(z) > 0$. On the contrary, when $z$ is *unrepresentative*, $\mathcal{M}(z) \approx 0$. Moreover, if the selected data point $z$ is *harmful (e.g. noisy samples)*, the retention of $z$ is a hindrance to the learning process on $S/z$, so the risk would be high; removing the harmful $z$ could result in a lower risk value. Hence, $\mathcal{M}(z) < 0$.

### 3.3    Gradient-based approximator

The exact calculation of MoSo, as shown in Eq.(3), has a quadratic time complexity of $\mathcal{O}(Tn^2)$, considering a dataset with $n$ samples and a total of $T$ training iterations required to obtain the surrogate network. However, this is practically infeasible; for instance, it may take more than 45 years to process the ImageNet-1K dataset using a Tesla-V100 GPU. To address this issue, we propose an efficient first-order approximator for calculating the MoSo score, which reduces the complexity to $\mathcal{O}(Tn)$. This approximation not only significantly decreases the computational requirements but also maintains the effectiveness of the MoSo score in practical applications.

**Proposition 1.1.** *The MoSo score could be efficiently approximated with linear complexity, that is,*

$$\hat{\mathcal{M}}(z) = \mathbb{E}_{t \sim \text{uniform}\{1,\ldots,T\}}\Big(\frac{T}{N}\eta_t \nabla \mathcal{L}(S/z, \mathbf{w}_t)^{\mathrm{T}} \nabla l(z, \mathbf{w}_t)\Big), \tag{4}$$

*where $S/z$ indicates the dataset $S$ excluding $z$, $l(\cdot)$ is the cross-entropy loss function and $\mathcal{L}(\cdot)$ means the average cross-entropy loss, $\nabla$ is the gradient operator with respect to the network parameters, and $\{(\mathbf{w}_t, \eta_t)|_{t=1}^{T}\}$ denotes a series of parameters and learning rate during training the surrogate network on $S$ with the SGD optimizer. $T$ is the maximum time-steps and $N$ is the training set size.*

The MoSo score approximator in Eq.(4) essentially represents the mathematical expectation of the inner product between the gradient with respect to network parameters considering only sample $z$ and the gradient using the training set excluding $z$ (denoted as $S/z$) over $T$ learning iterations. A sample $z$ will be assigned a higher MoSo score if the mathematical expectation of the inner product is larger. This can be intuitively understood as follows: if the gradient $\nabla l(z, \mathbf{w})$ from sample $z$ is consistent with the average gradient vector $\nabla \mathcal{L}(S/z, \mathbf{w})$, it implies that optimizing the network using sample $z$ will yield a similar effect on reducing the empirical risk as using all remaining samples. This indicates that sample $z$ is an important and representative sample. Concurrently, according to Eq.(4), it is also assigned a high MoSo score, which aligns with the intuition.

### 3.4    Theoretical analysis of tightness and complexity

First, we provide a rigorous mathematical justification to show that there is a theoretical guarantee for the error between the approximator we provide and the exact score by the brute-force leave-one-out retraining.

**Proposition 1.2.** *By supposing the loss function is $\ell$-Lipschitz continuous and the gradient norm of the network parameter is upper-bounded by $g$, and setting the learning rate as a constant $\eta$, the approximation error of Eq. (4) is bounded by:*

$$|\mathcal{M}(z) - \hat{\mathcal{M}}(z)| \leqslant \mathcal{O}\Big((\ell\eta + 1)gT + \eta g^2 T\Big), \tag{5}$$

*where $T$ is the maximum iterations.*

The proposition shows that approximation error is positively correlated with many factors such as training duration $T$, gradient norm $g$, and learning rate $\eta$. In order to control the impact of approximate errors, in practice, we will not train the surrogate network to complete convergence, instead, we will only update a small number of epochs.

**Complexity analysis.** We show that Eq.(4) efficiently approximates the MoSo score with linear complexity. Specifically, calculating the overall gradient information requires a time complexity of $\mathcal{O}(n)$. Additionally, computing the expectation of the gradient over different training iterations in Eq.(4) takes $T$ steps, resulting in a total complexity of $\mathcal{O}(Tn)$. In practice, we can randomly sample a few time steps rather than considering all $T$ steps to calculate the mathematical expectation, reducing the overall complexity to be less than $\mathcal{O}(Tn)$. Moreover, the optimized gradient calculation operator,

**Algorithm 1:** Data Pruning with MoSo.

---

**Require:** Dataset $\mathcal{S} = \{(x_i, y_i)|_{i=1}^N\}$, pruning ratio $\delta$;
**Require:** Random initialized the parameter $\mathbf{w}_0$ of a network;
**Require:** cross-entropy loss $l(\cdot)$, SGD optimizer;
**Require:** Learning-rate scheduler $\{\eta_1, ..., \eta_T\}$, maximum iteration $T$;

1: Initialize the sample-wise score set $\mathcal{V} = \phi$ as a null set;
2: **if** multiple computing devices are available **then**
3:    Partitioning $\mathcal{S}$ into $\mathbb{S} : \{S_1, ..., S_I\}$;                    //With parallel acceleration.
4: **else**
5:    $\mathbb{S} = \{\mathcal{S}\}$                                          //Without acceleration.
6: **end if**
7: **for** $S_i \in \mathbb{S}$ **do**
8:    $\{(\mathbf{w}_t, \eta_t)|_{t=1}^T\} = \text{SGD}\Big(\mathcal{L}(S_i, \mathbf{w}_0), \ T, \ \{\eta_1, ..., \eta_T\}\Big)$;          //Train the surrogate network.
9:    **for** $z \in S_i$ **do**
10:       $\mathcal{M}(z) \leftarrow \mathbb{E}_{t \sim \text{uniform}\{1,...,T\}}\Big(\eta_t \nabla \mathcal{L}(\mathcal{S}/z, \mathbf{w}_t)^\text{T} \nabla l(z, \mathbf{w}_t)\Big)$;          //MoSo scoring.
11:       Merge into the score set $\mathcal{V} \leftarrow \mathcal{V} + \{\mathcal{M}(z)\}$
12:    **end for**
13: **end for**
14: **return** $\widehat{S} \leftarrow \text{Pruning}(\mathcal{S}|\mathcal{V}, \delta)$                              //Data pruning.

---

implemented by advanced deep learning frameworks such as PyTorch [2], further accelerates the computation, making it more feasible for practical applications.

### 3.5 Data Pruning with MoSo

This section presents the pipeline for utilizing the approximated MoSo score in data pruning and coreset selection. The pseudocode is provided in Algorithm 1. In the **MoSo scoring** step (see line 10 of Algorithm 1), we employ Eq.(4) from Proposition 1 to calculate the MoSo score. In practice, there is no need to sum all the time steps $\{1, ..., T\}$ when calculating the mathematical expectation. Instead, an efficient approach is randomly sampling several time steps for calculating the average or expectation, reducing the overall computational cost. In the **data pruning** step, samples with low scores are pruned, and the proportion of pruned data corresponds to the predefined ratio $\delta$.

**Parallel acceleration by dataset partitioning.** To enhance the practical applicability of MoSo on large-scale datasets, we propose a parallel acceleration scheme that can be employed when multiple GPU devices are available. Specifically, before training the surrogate network, we initially divide the original full dataset into a series of **non-overlapping** subsets. This approach enables efficient processing by leveraging the parallel computing capabilities of multiple GPUs, where the number of subsets should be no more than the available computing devices. We select a subset from $\mathbb{S}$ without replacement for each device and then perform training and scoring within the chosen subset, following Algorithm 1. Finally, MoSo scores from different devices are combined together. As long as the number of samples in a subset is large enough to approximately represent the overall statistics of the dataset, this partitioning scheme will not compromise performance while significantly reducing computation time by a factor of $I$. This approach is particularly useful for large-scale datasets.

## 4 Experiments

**Datasets and baselines.** We evaluate our method on three well-known public benchmarks: the CIFAR-100 [5], which contains 50,000 training examples of 100 categories; the Tiny-ImageNet [49], which has 100,000 images of 200 classes; and the ImageNet-1K [31], which covers 1000 classes with more than 1M training images. We compare our method with a range of baselines, including: (1). Random selection; (2). Herding [29]; (3). Forgetting [28]; (4). GraNd [27]; (5). EL2N [27];

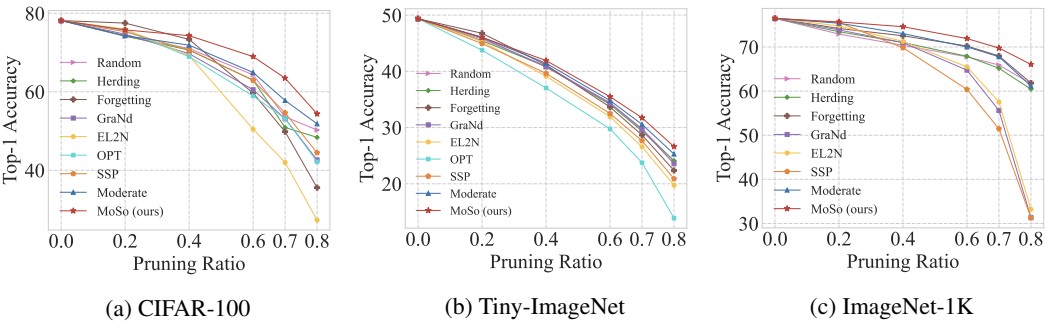

(a) CIFAR-100      (b) Tiny-ImageNet      (c) ImageNet-1K

Figure 1: Performance comparison of our proposed MoSo and other baseline methods on three image classification datasets: CIFAR-100 [5], Tiny-ImageNet [49], and ImageNet-1K [31]. The results show that our approach outperforms most of the baselines, especially for the high pruning rate (e.g., 70%, 80%).

(6). Optimization-based Dataset Pruning (OPT) [38]; (7). Self-supervised pruning (SSP) [42]; (8). Moderate [48].

**Implementation details.** We implement our method in Pytorch [2]. All the experiments are run on a server with 8 Tesla-V100 GPUs. Unless otherwise specified, we use the same network structure ResNet-50 [22] for both the coreset and the surrogate network on the full data. We keep all hyper-parameters and experimental settings of training before and after dataset pruning consistent. We train the surrogate network on all datasets for 50 epochs. To estimate the mathematical expectation in Eq.(4) from Proposition 1, we randomly sample 10 time steps. Thus, MoSo can compute gradients from multiple epochs without increasing the overall time cost significantly, compared to methods that need to train a network fully (*e.g.* 200 epochs for CIFAR-100) before calculating the scores.

## 4.1 Main Results

In the following subsections, we present the detailed performance comparison of our method and baselines on three experiments: data pruning, generalization to unseen structures, and robustness to label noise.

**Data pruning.** As Figure 1(a) and 1(b) show, our method significantly surpasses the SOTA method [48] on CIFAR-100 and Tiny-ImageNet at high pruning ratios. Note that some selected baselines perform worse than the random selection, especially when with high pruning ratios, *e.g.* the results of the classic EL2N on CIFAR-100, while our method doesn't suffer from this problem. Furthermore, in Figure 1(c), our method achieves satisfactory performances on ImageNet-1K across different pruning rates, showing its effectiveness on large-scale and complex datasets. These results also indicate that our method can capture the sample importance more accurately and robustly than the existing methods.

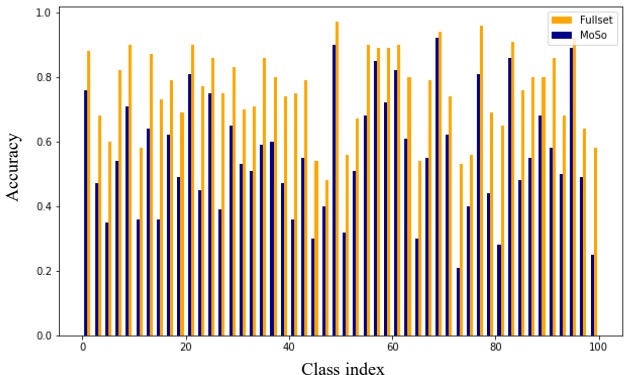

Figure 2: We show the class-wise accuracy before (bars named *Fullset*) and after (bars named *MoSo*) applying our MoSo approach. The experiment is conducted on CIFAR-100. We chose ResNet-18 as the network architecture and set the pruning ratio to be 0.8.

To study whether our algorithm improves/hurts certain classes, we visualize the class-wise accuracy before and after applying our MoSo data pruning approach in Figure 2. We observe a significant correlation between the two, with a Spearman correlation coefficient of 0.913 and a P value of

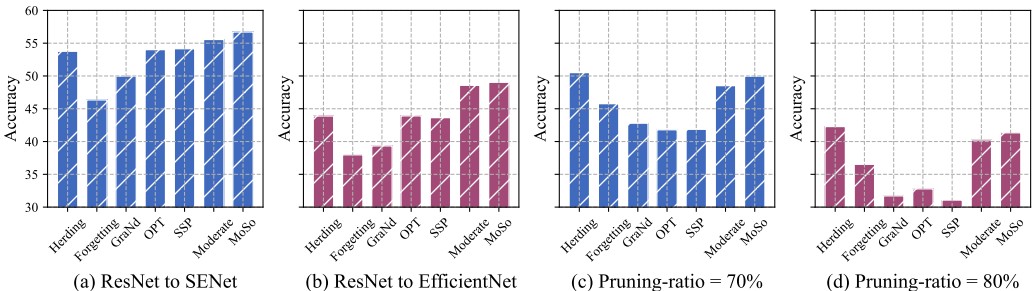

| (a) ResNet to SENet | (b) ResNet to EfficientNet | (c) Pruning-ratio = 70% | (d) Pruning-ratio = 80% |

Figure 3: In (a) and (b), we study the generalization performance of MoSo and other baselines on CIFAR-100 from ResNet-50 to SENet (R to S) and ResNet-50 to EfficientNet-B0 (R to E). In (c) and (d), we show the robustness against label noise of MoSo and other baselines on CIFAR-100, where the labels are randomly replaced with any possible label with a 20% probability.

0.0295. This indicates that the performance before and after pruning with MoSo is consistent across categories, and no significant improvement or harm to any particular category is observed.

**Generalization test.** To test whether the pruning results are overfitting to the specific network architecture, we evaluate MoSo's generalization ability to unseen architectures. Following the protocol in [48], we use ResNet-50 as the surrogate network for scoring, and training different network architectures, SENet [18] and EfficientNet-B0 [30], on the selected data. Figure 3(a) and Figure 3(b) show the experimental results on different network architectures. MoSo exhibits a satisfying generalization ability to unseen models and consistently outperforms or matches the state-of-the-art methods such as SSP [42], and OPT [38].

**Robustness test.** Label noise is a common challenge in real-world applications. Therefore, how to improve the model robustness to label noise is an important and popular problem. In this section, we study the robustness of MoSo to label noise by conducting comparative experiments on CIFAR-100 and Tiny-ImageNet with synthetic label noise. Specifically, we inject label noise [14] into the two datasets by randomly replacing the labels for a percentage of the training data with all possible labels. The noise rate is set to 20% for all the experiments. We use ResNet-50 as the network architecture and keep all experimental settings consistent with the previous data pruning experiments. The results are shown in Figure 3(c) and Figure 3(d). We observe that some difficulty-based importance criteria don't work very well, such as Forgetting, GraNd on both settings, while Herding [29], Moderate [48] and our MoSo, perform well and significantly outperform other baselines in all settings. Our MoSo achieves comparable performance with the best baseline, Herding [29], only lagging behind Herding by less than 1% Top-1 acc.

## 4.2 Computational Efficiency

We evaluated MoSo and the other baseline methods on a server with 8 Tesla V100 GPUs. We used the CIFAR-100 dataset and the ResNet50 backbone for our experiments. It should be noted here that we also take into account the training time of the surrogate network. This is because not all datasets will have a community-provided network to calculate scores, and private datasets will require practitioners to train a surrogate network. MoSo achieves the best trade-off between computational requirements and performance, making it the best-performing model with rea-

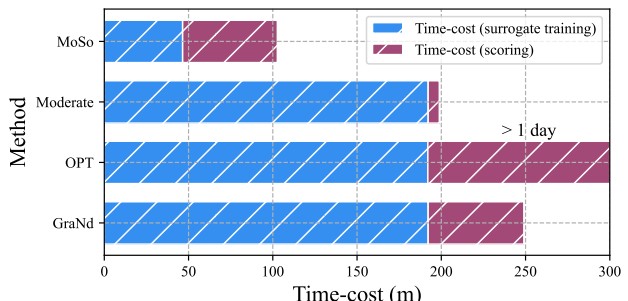

Figure 4: Time-core comparison between our MoSo and other baselines. Please note that when implementing the GraNd method, we don't take the summation of the gradient norm from all epochs, instead, we use the same time-step sampling scheme as MoSo.

sonable computational demands. Notably, it outperforms the state-of-the-art method, Moderate, while being more efficient. Because of the use of large-scale linear programming in the scoring phase, OPT is significantly more time-consuming than the other methods.

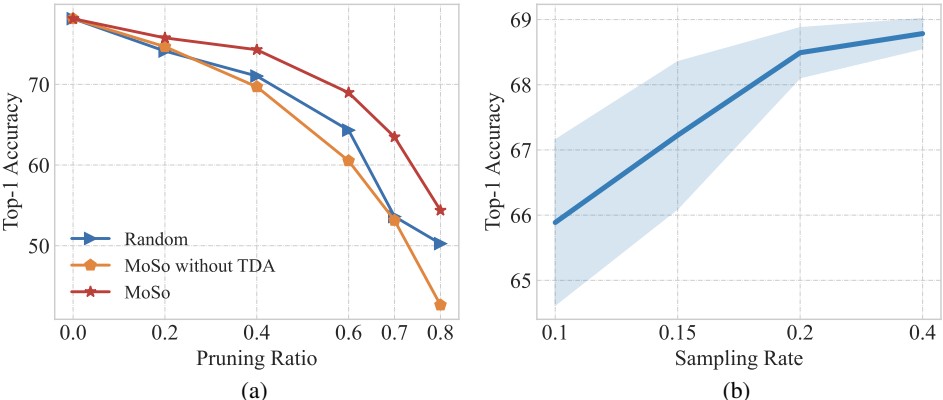

Figure 5: Ablation study on the effect of (a) incorporating the training-dynamic-awareness (TDA) into the MoSo score, and (b) using different time step sampling rates on the accuracy of the selected coreset (with a pruning ratio of 0.4). The experiments are conducted on CIFAR-100 with ResNet-50 as the network architecture.

## 4.3 Further Study

In this subsection, we perform additional ablation experiments to investigate the effect of the awareness of training dynamics, the effect of time step sampling, the effect of the parallel speed-up scheme (dataset partitioning), and the effect of the number of epochs in the surrogate training stage.

**Effect of the awareness of training dynamics.** Here we investigate the effect of incorporating the awareness of training dynamics into our MoSo score. To do this, we compare our method with a variant that removes this awareness by only considering the gradient from the very last epoch of the surrogate network. This variant is equivalent to using the gradient norm as a measure of sample importance. The results are shown in Figure 5(a). We can clearly see that our method outperforms the variant on both CIFAR-100 across different pruning rates. This indicates that the awareness of training dynamics is crucial for capturing the impact of a sample on the model performance and that the gradient norm alone from a converged network is not sufficient for measuring sample importance.

**Effect of time step sampling.** We then investigate the effect of time step sampling on the accuracy and efficiency of our method. Time step sampling is a technique that we use to reduce the computational cost of calculating Eq.(4) by randomly selecting a subset of epochs to estimate the MoSo score. However, this technique also introduces variance into the estimation and may even affect the quality of the selected coreset with a too-small sampling rate. To study this trade-off, we conduct experiments with different sampling rates and measure the performance of the final coreset on CIFAR-100. The results are shown in Figure 5(b). As expected, we observe that the mean performance decreases as the sampling rate decreases, and the variance also increases as the sampling rate decreases. This suggests that time-step sampling is a useful technique for improving the efficiency of our method, but it should be used with caution to avoid sacrificing too much accuracy.

**Effect of the parallel speed-up scheme.** The most time-consuming aspect of data pruning is training the surrogate network. Our framework utilizes a parallel speed-up scheme, explained in line 3 of Algorithm 1 in the main text. Essentially, we partition the original full set into several non-overlapping subsets with equivalent size $\mathcal{S} \rightarrow \{S_1, ..., S_I\}$, where $I$ represents the number of computing devices. On each device,

Table 1: The effect of dataset partitioning on the final data pruning performance, where the pruning ratio is $0.2$. The bold one represents the settings used in this work.

| Subsets number $I$ | 1 | 2 | **5** | 10 |
|---|---|---|---|---|
| MoSo (ours) | 74.35 | 75.11 | 75.76 | 75.81 |

we can train a surrogate network on $S_i$. Then, for each sample $z \in S_i$, we perform MoSo scoring within the current set by using $\mathcal{M}(z|S_i)$ to approximate $\mathcal{M}(z|\mathcal{S})$. Implementing this approach reduces the overall time overhead by $I$ fold. Table 1 demonstrates that partitioning $\mathcal{S}$ into more subsets can improve data pruning's performance. This means that if the training set is vast, a single sample's impact may be buried, making it challenging to measure. In a relatively small training set, however, the effect of a single sample can be more sensitively reflected and easily captured.

**Effect of the number of epochs in the surrogate training stage.** To investigate the impact of surrogate network training epochs on the ultimate data pruning performance, we augmented the number of training epochs and presented the experimental outcomes on CIFAR-100, as shown in Table 2. However, it is evident that augmenting the training duration does not lead to a uniform improvement in performance. For instance, lengthening the train-

Table 2: The effect of surrogate network training epochs on the final data pruning performance, where the pruning ratio is 0.2.

| Training epochs | 50 | 100 | 150 | 200 |
|---|---|---|---|---|
| MoSo | 75.76 | 76.41 | 76.19 | 76.58 |

ing duration from 50 epochs to 200 epochs only resulted in a meager 0.82 Top-1 accuracy gain, while the time consumed quadrupled. Note that this is consistent with our conclusion in Proposition 1.2 that the approximation error between the approximated MoSo and the exact MoSo value by leave-one-out-retraining increases with time (T), leading to no improvement in DP performance with longer training time.

## 5 Conclusion

This paper introduces a novel metric for measuring sample importance, called the Moving-one-Sample-out (MoSo) score. It quantifies sample importance by measuring the change of the optimal empirical risk when a specific sample is removed from the training set. By doing so, MoSo can better distinguish important samples that contribute to the model performance from harmful noise samples that degrade it, as the former tends to lower the empirical risk, while the latter may increase it. Moreover, we propose an efficient estimator for MoSo with linear complexity and approximation error guarantees. The estimator incorporates the awareness of training dynamics by considering the gradient difference across different epochs. We conduct extensive experiments on various data pruning tasks and demonstrate the effectiveness and generalization of our method.

**Limitations and future work.** First, the MoSo score is actually the agreement between the gradient of a single sample and the mathematical expectation of the gradient. The higher the agreement, the sample will be given a higher score. In fact, this is based on the important assumption that the amount of information in the data set is much greater than the amount of noise. However, if the amount of noise is dominant, the usefulness of MoSo is not guaranteed. Therefore, we believe that in the future, it is very necessary to propose a variant of MoSo to adapt to high noise conditions with theoretical guarantees. Secondly, in terms of application, this paper only evaluates the performance of MoSo on classification tasks. Many practical tasks, *e.g.* large-scale multimodal learning, are worth considering in future work.

**Social Impact.** MoSo has potential influences in important applications such as data collection and data-efficient AI. Moreover, it is beneficial for reducing the computational workload during training and the cost of storing datasets, which is of great significance for environmentally friendly and energy-friendly economies. But it may be deployed for inhumane web-data monitoring. The potential negative effects can be avoided by implementing strict and secure data privacy regulations.

## 6 Acknowledgment

This work has been supported by Hong Kong Research Grant Council - Early Career Scheme (Grant No. 27209621), General Research Fund Scheme (Grant No. 17202422), Theme-based Research (T45-702/22-R), and RGC Matching Fund Scheme (RMGS). Part of the described research work is conducted in the JC STEM Lab of Robotics for Soft Materials funded by The Hong Kong Jockey Club Charities Trust.

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
