# Supplementary Material for MoSo

**Haoru Tan**[1,3] *    **Sitong Wu**[2,3] *    **Fei Du**[3,4]

**Yukang Chen**[2]    **Zhibin Wang**[3,4]    **Fan Wang**[3,4]    **Xiaojuan Qi**[1]

[1]HKU    [2]CUHK    [3]DAMO Academy, Alibaba Group    [4]Hupan Lab, Zhejiang Province

## 1  Mathematical Proof

Before the proof, we first revisit the definition of MoSo.

**Definition 1.** *The MoSo score for a specific sample $z$ selected from the training set $\mathcal{S}$ is*

$$\mathcal{M}(z) = \mathcal{L}\Big(\mathcal{S}/z, \mathbf{w}^*_{\mathcal{S}/z}\Big) - \mathcal{L}\Big(\mathcal{S}/z, \mathbf{w}^*_{\mathcal{S}}\Big), \tag{1}$$

*where $\mathcal{S}/z$ indicates the dataset $\mathcal{S}$ excluding $z$, $\mathcal{L}(\cdot)$ is the average cross-entropy loss on the considered set of samples, $\mathbf{w}^*_{\mathcal{S}}$ is the optimal parameter trained on the full set $\mathcal{S}$, and $\mathbf{w}^*_{\mathcal{S}/z}$ is the optimal parameter on $\mathcal{S}/z$.*

### 1.1  Proof for Proposition 1.1

**Proposition 1.1.** *The MoSo score could be efficiently approximated with linear complexity, that is,*

$$\hat{\mathcal{M}}(z) = \mathbb{E}_{t\sim\text{uniform}\{1,\dots,T\}}\Big(\eta_t \nabla\mathcal{L}(\mathcal{S}/z, \mathbf{w}_t)^{\mathrm{T}} \nabla l(z, \mathbf{w}_t)\Big), \tag{2}$$

*where $\mathcal{S}/z$ indicates the dataset $\mathcal{S}$ excluding $z$, $l(\cdot)$ is the cross-entropy loss function and $\mathcal{L}(\cdot)$ means the average cross-entropy loss, $\nabla$ is the gradient operator with respect to the network parameters, and $\{(\mathbf{w}_t, \eta_t)|_{t=1}^{T}\}$ denotes a series of parameters and learning rate during training the surrogate network on $\mathcal{S}$ with the SGD optimizer.*

**Proof.**

Given a specific sample $z$, we present a unified loss formulation:

$$\mathcal{L}_\epsilon = \frac{1}{N}\sum_{(x,y)\in\mathcal{L}}^{N} l\Big[(x,y), \mathbf{w}\Big] + \epsilon\cdot l\Big[z, \mathbf{w}\Big], \tag{3}$$

where $\epsilon$ is a coefficient. Hence, we have $\mathcal{L}(\mathcal{S}, \mathbf{w}) = \mathcal{L}_{\epsilon:0}$ and $\mathcal{L}(\mathcal{S}/z, \mathbf{w}) = \mathcal{L}_{\epsilon:\frac{-1}{N}}$. We suppose that, with the SGD optimizer, the training process reaches the optimal solution after $T$ steps,

$$\mathbf{w}^* = \mathbf{w}^T_{\mathcal{S}} = \arg\min \mathcal{L}_{\epsilon:0}, \quad \mathbf{w}^*_{\mathcal{S}/z} = \mathbf{w}^T_{\mathcal{S}/z} = \arg\min \mathcal{L}_{\epsilon:\frac{-1}{N}}. \tag{4}$$

where $\mathbf{w}^* = \mathbf{w}^T_{\mathcal{S}}$ and $\mathbf{w}^t = \mathbf{w}^t_{\mathcal{S}}$ for simplicity.

Hence, the MoSo-score could be re-writed as:

$$\mathcal{M}(z) = \mathcal{L}^T_{\epsilon:\frac{-1}{N}} - \mathcal{L}^T_{\epsilon:0} + \frac{1}{N}\cdot l\Big(z, \mathbf{w}^T_{\mathcal{S}}\Big),$$

---

*Equal contribution.

37th Conference on Neural Information Processing Systems (NeurIPS 2023).

and, we use $\mathcal{M}^t(z)$ to denote the empirical risk on $\mathcal{S}/z$ gap at the $t$-th step,

$$\mathcal{M}^t(z) = \mathcal{L}^t_{\epsilon:\frac{-1}{N}} - \mathcal{L}^t_{\epsilon:0} + \frac{1}{N} \cdot l\Big(z, \mathbf{w}^t_{\mathcal{S}}\Big),$$

where $t \leqslant T$. We use $\mathcal{M}(z)$ to denote $\mathcal{M}^T(z)$. Note that the network on the full set $\mathcal{S}$ and that on the subset $\mathcal{S}/z$ is started from the same initialization, that is, $\mathcal{M}^0(z) = 0$. Let's start with the identical equation below,

$$
\begin{aligned}
\mathcal{M}(z) &= \Big(\mathcal{M}(z) - \mathcal{M}^{T-1}(z)\Big) + \Big(\mathcal{M}^{T-1}(z) - \mathcal{M}^{T-2}(z)\Big) + ... + \Big(\mathcal{M}^1(z) - \mathcal{M}^0(z)\Big) + \mathcal{M}^0(z) \\
&= \Big(\mathcal{M}(z) - \mathcal{M}^{T-1}(z)\Big) + \Big(\mathcal{M}^{T-1}(z) - \mathcal{M}^{T-2}(z)\Big) + ... + \Big(\mathcal{M}^1(z) - \mathcal{M}^0(z)\Big) \\
&= \Delta\mathcal{M}^T + \Delta\mathcal{M}^{T-1} + ... + \Delta\mathcal{M}^1.
\end{aligned}
\tag{5}
$$

Let's take one single item $\Delta\mathcal{M}^t$ as an example,

$$
\begin{aligned}
\Delta\mathcal{M}^t &= \mathcal{M}^t(z) - \mathcal{M}^{t-1}(z) \\
&= \Big[\mathcal{L}^t_{\epsilon:\frac{-1}{N}} - \mathcal{L}^t_{\epsilon:0} + \frac{1}{N}l\Big(z, \mathbf{w}^t_{\mathcal{S}}\Big)\Big] - \Big[\mathcal{L}^{t-1}_{\epsilon:\frac{-1}{N}} - \mathcal{L}^{t-1}_{\epsilon:0} + \frac{1}{N}l\Big(z, \mathbf{w}^{t-1}_{\mathcal{S}}\Big)\Big] \\
&= \Big[\mathcal{L}^t_{\epsilon:\frac{-1}{N}} - \mathcal{L}^{t-1}_{\epsilon:\frac{-1}{N}}\Big] - \Big[\mathcal{L}^t_{\epsilon:0} - \mathcal{L}^{t-1}_{\epsilon:0}\Big] + \frac{1}{N}\Big[l\Big(z, \mathbf{w}^t_{\mathcal{S}}\Big) - l\Big(z, \mathbf{w}^{t-1}_{\mathcal{S}}\Big)\Big].
\end{aligned}
\tag{6}
$$

By using the first-order Taylor approximation to approximate $\mathcal{L}^t$ with $\mathcal{L}^{t-1}$, we estimate $\Delta\mathcal{M}^t$ with,

$$\widehat{\Delta\mathcal{M}^t} = [\nabla\mathcal{L}^{t-1}_{\epsilon:\frac{-1}{N}}]^{\mathrm{T}}\Big(\mathbf{w}^t_{\mathcal{S}/z} - \mathbf{w}^{t-1}_{\mathcal{S}/z}\Big) - [\nabla\mathcal{L}^{t-1}_{\epsilon:0}]^{\mathrm{T}}\Big(\mathbf{w}^t_{\mathcal{S}} - \mathbf{w}^{t-1}_{\mathcal{S}}\Big) + \frac{1}{N}\nabla l\Big(z, \mathbf{w}^{t-1}_{\mathcal{S}}\Big)^{\mathrm{T}}\Big(\mathbf{w}^t_{\mathcal{S}} - \mathbf{w}^{t-1}_{\mathcal{S}}\Big).
\tag{7}$$

According to the update rule of the SGD optimizer, that is, $\mathbf{w}^t = \mathbf{w}^{t-1} - \eta_t\nabla\mathcal{L}^{t-1}$, $\Delta\mathcal{M}^t$ could be converted into

$$\widehat{\Delta\mathcal{M}^t} = -\eta_t||\nabla\mathcal{L}^t_{\epsilon:\frac{-1}{N}}||^2 + \eta_t||\nabla\mathcal{L}^{t-1}_{\epsilon:0}||^2 - \eta_t\frac{1}{N}\nabla l\Big(z, \mathbf{w}^{t-1}_{\mathcal{S}}\Big)^{\mathrm{T}}\nabla\mathcal{L}^{t-1}_{\epsilon:0}.
\tag{8}$$

Here, we use the Taylor approximation again to approximate $\nabla\mathcal{L}^{t-1}_{\epsilon:\frac{-1}{N}}$ with $\nabla\mathcal{L}^{t-1}_{\epsilon:0}$,

$$
\begin{aligned}
\nabla\mathcal{L}^{t-1}_{\epsilon:\frac{-1}{N}} &\approx \nabla\mathcal{L}^{t-1}_{\epsilon:0} + \frac{\partial\mathcal{L}^{t-1}}{\partial\epsilon}|_{\epsilon=0}\Big((\epsilon:\frac{-1}{N}) - (\epsilon:0)\Big) \\
&= \nabla\mathcal{L}^{t-1}_{\epsilon:0} - \frac{1}{N}\frac{\partial\nabla\mathcal{L}^{t-1}}{\partial\epsilon}|_{\epsilon=0},
\end{aligned}
\tag{9}
$$

where $\frac{\partial\nabla\mathcal{L}^{t-1}}{\partial\epsilon}|_{\epsilon=0} = \nabla l(z, \mathbf{w}^{t-1}_{\mathcal{S}})$. By substituting Eq. (9) into Eq. (8), we have that,

$$
\begin{aligned}
\widetilde{\Delta\mathcal{M}^t} &= \frac{\eta_t}{N}\Big[\nabla\mathcal{L}^{t-1}_{\epsilon:0} - \frac{1}{N}\nabla l\Big(z, \mathbf{w}^{t-1}_{\mathcal{S}}\Big)\Big]^{\mathrm{T}}\nabla l\Big(z, \mathbf{w}^{t-1}_{\mathcal{S}}\Big) \\
&= \frac{\eta_t}{N}\nabla\mathcal{L}(\mathcal{S}/z, \mathbf{w}^{t-1}_{\mathcal{S}})^{\mathrm{T}}\nabla l(z, \mathbf{w}^{t-1}_{\mathcal{S}}).
\end{aligned}
\tag{10}
$$

By substituting Eq. (10) into Eq. (5), we have that,

$$
\begin{aligned}
\mathcal{M}(z) &= \Delta\mathcal{M}^T + \Delta\mathcal{M}^{T-1} + ... + \Delta\mathcal{M}^1 \\
&\approx \widetilde{\Delta\mathcal{M}^T} + \widetilde{\Delta\mathcal{M}^{T-1}} + ... + \widetilde{\Delta\mathcal{M}^1} \\
&= \sum_t \frac{\eta_t}{N}\nabla\mathcal{L}(\mathcal{S}/z, \mathbf{w}_t)^{\mathrm{T}}\nabla l(z, \mathbf{w}_t), \\
&= \frac{T}{N}\sum_t \frac{\eta_t}{T}\nabla\mathcal{L}(\mathcal{S}/z, \mathbf{w}_t)^{\mathrm{T}}\nabla l(z, \mathbf{w}_t) \\
&= \frac{T}{N} \cdot \mathbb{E}_{t\sim\text{uniform}\{1,...,T\}}\Big(\eta_t\nabla\mathcal{L}(\mathcal{S}/z, \mathbf{w}_t)^{\mathrm{T}}\nabla l(z, \mathbf{w}_t)\Big).
\end{aligned}
\tag{11}
$$

In practice, $\frac{T}{N}$ is just a constant that contributes little, where $N$ is the number of all training data and $T$ is the number of update steps in training. Moreover, sometimes numerical instability may occur due to factors such as $N$ or $T$ being too large, so we completely ignore this insignificant constant in our applications. Thus, we have the final approximator,

$$\hat{\mathcal{M}}(z) = \mathbb{E}_{t \sim \text{uniform}\{1,\ldots,T\}} \Big( \eta_t \nabla \mathcal{L}(\mathcal{S}/z, \mathbf{w}_t)^{\mathrm{T}} \nabla l(z, \mathbf{w}_t) \Big).$$

So, Proposition 1.1 has been proven.

## 1.2 Proof for Proposition 1.2

**Proposition 1.2.** *By supposing the loss function is $\ell$-Lipschitz continuous and the gradient norm of the network parameter is upper-bounded by $g$, and setting the learning rate as a constant $\eta$, the approximation error of Eq. (2) is bounded by:*

$$|\mathcal{M}(z) - \hat{\mathcal{M}}(z)| \leqslant \mathcal{O}\Big( (\ell\eta + 1)gT + \eta g^2 T \Big), \tag{12}$$

*where $T$ is the maximum iteration.*

### 1.2.1 Proof for Proposition 1.2.

Note that the final approximator is the time domain mathematical expectation for $\Delta\widetilde{\mathcal{M}^t}$, which is used to replace the untraceable $\Delta\mathcal{M}^t$, so we analyze the overall error by starting from $|\Delta\mathcal{M}^t - \Delta\widetilde{\mathcal{M}^t}|$,

$$|\Delta\mathcal{M}^t - \Delta\widetilde{\mathcal{M}^t}| \leqslant |\Delta\mathcal{M}^t - \Delta\widehat{\mathcal{M}^t}| + |\Delta\widehat{\mathcal{M}^t} - \Delta\widetilde{\mathcal{M}^t}|,$$

where the first $|\Delta\mathcal{M}^t - \Delta\widehat{\mathcal{M}^t}|$ occurs when approximating $\mathcal{L}^t$ with $\mathcal{L}^{t-1}$ in Eq.(7), the other one occurs when approximating $\nabla\mathcal{L}_{\epsilon:\frac{-1}{N}}^{t-1}$ with $\nabla\mathcal{L}_{\epsilon:0}^{t-1}$ in Eq.(9).

As for the first approximation error,

$$
\begin{aligned}
\mathcal{O}(|\Delta\mathcal{M}^t - \Delta\widehat{\mathcal{M}^t}|) \;\; &\propto \mathcal{O}(|\mathcal{L}^t - \widehat{\mathcal{L}^t}|) \\
&= \mathcal{O}(|\mathcal{L}^t - \mathcal{L}^{t-1} - \nabla\mathcal{L}^{t-1}(\mathbf{w}^t - \mathbf{w}^{t-1})|) \\
&\leqslant \mathcal{O}(|\mathcal{L}^t - \mathcal{L}^{t-1}| + |\nabla\mathcal{L}^{t-1}(\mathbf{w}^t - \mathbf{w}^{t-1})|),
\end{aligned}
\tag{13}
$$

since the loss function is $\ell$-Lipschitz continuous by the mild assumption, we have that,

$$\mathcal{O}(|\mathcal{L}^t - \mathcal{L}^{t-1}| + |\nabla\mathcal{L}^{t-1}(\mathbf{w}^t - \mathbf{w}^{t-1})|) \leqslant \mathcal{O}(\ell|\mathbf{w}^t - \mathbf{w}^{t-1}| + |\nabla\mathcal{L}^{t-1}(\mathbf{w}^t - \mathbf{w}^{t-1})|), \tag{14}$$

according to the update rule in SGD, we have $\mathbf{w}^t = \mathbf{w}^{t-1} - \eta\nabla\mathcal{L}^{t-1}$, so,

$$\mathcal{O}(|\Delta\mathcal{M}^t - \Delta\widehat{\mathcal{M}^t}|) \leqslant \mathcal{O}(\ell\eta|\nabla\mathcal{L}^{t-1}| + \eta||\nabla\mathcal{L}^{t-1}||^2). \tag{15}$$

Since the gradient norm is upper-bounded by a constant $g$, thus,

$$\mathcal{O}(|\Delta\mathcal{M}^t - \Delta\widehat{\mathcal{M}^t}|) \leqslant \mathcal{O}(\ell\eta g + \eta g^2). \tag{16}$$

As for the second approximation error term $\mathcal{O}(|\Delta\widehat{\mathcal{M}^t} - \Delta\widetilde{\mathcal{M}^t}|)$, since it estimates $\nabla\mathcal{L}_{\epsilon:\frac{-1}{N}}^{t-1}$ with $\nabla\mathcal{L}_{\epsilon:0}^{t-1}$ in Eq.(9), we have that,

$$
\begin{aligned}
\mathcal{O}(|\Delta\widehat{\mathcal{M}^t} - \Delta\widetilde{\mathcal{M}^t}|) &\propto \;\; |\nabla\mathcal{L}_{\epsilon:\frac{-1}{N}}^{t-1} - \nabla\widehat{\mathcal{L}_{\epsilon:\frac{-1}{N}}^{t-1}}| \\
&= |\nabla\mathcal{L}_{\epsilon:\frac{-1}{N}}^{t-1} - (\nabla\mathcal{L}_{\epsilon:0}^{t-1} - \frac{1}{N}\frac{\partial\nabla\mathcal{L}^{t-1}}{\partial\epsilon}|_{\epsilon=0})| \\
&\leqslant |\nabla\mathcal{L}_{\epsilon:\frac{-1}{N}}^{t-1}| + |\nabla\mathcal{L}_{\epsilon:0}^{t-1}| + \Big|\frac{1}{N}\frac{\partial\nabla\mathcal{L}^{t-1}}{\partial\epsilon}|_{\epsilon=0}\Big|,
\end{aligned}
\tag{17}
$$

where $\frac{\partial\nabla\mathcal{L}^{t-1}}{\partial\epsilon}|_{\epsilon=0} = \nabla l(z, \mathbf{w}_{\mathcal{S}}^{t-1})$. Since the gradient norm is bounded by constant $g$ and $N$ is generally a quite big value (e.g., $N = 1M$ for ImageNet), so,

$$|\nabla\mathcal{L}_{\epsilon:\frac{-1}{N}}^{t-1}| + |\nabla\mathcal{L}_{\epsilon:0}^{t-1}| + \Big|\frac{1}{N}\frac{\partial\nabla\mathcal{L}^{t-1}}{\partial\epsilon}|_{\epsilon=0}\Big| \approx \mathcal{O}(g). \tag{18}$$

By jointly considering Eq.(16) and Eq.(18) and then taking the summation from $t = 1$ to $T$, we have that,

$$\mathcal{O}(|\mathcal{M}(z) - \hat{\mathcal{M}}(z)|) \leqslant \mathcal{O}\left(\ell \eta g T + \eta g^2 T + g T\right) = \mathcal{O}\left((\ell \eta + 1) g T + \eta g^2 T\right).$$

Proposition 1.2 has been proven.