# OpenReview forum: "Data Pruning via Moving-one-Sample-out"
_NeurIPS.cc/2023/Conference — NeurIPS 2023 poster_

### Official Review · Reviewer_DWe5 · 2023-06-30

**Soundness:** 3 good
**Presentation:** 3 good
**Contribution:** 3 good
**Rating:** 6
**Confidence:** 4

**Summary:**

The paper proposes a new method called moving-on-sample-out (MoSo) to remove less informative samples from the training data. The criteria for removed samples is based on the change in the optimal empirical risk when the sample is removed. As exact calculation of the criterion is computationally challenging, the authors propose an estimator based on gradient information. MoSo shows empirical success compared to baseline methods in data pruning, generalization between networks, and robustness to label noise.

**Strengths:**

- The method is based on a simple yet effective idea.
- The computational complexity problem of MoSo is effectively tackled with the proposed estimator.
- In the experimental section a wide range of comparison methods are used as baselines.
- An ablation study is performed to detail the necessity of sub-elements of the method.
- The paper is fairly easy to follow.
- MoSo is shown to outperform baseline methods in data pruning, generalization between networks, and robustness to label noise in terms of accuracy.

**Weaknesses:**

Main points:
- The justification for the gradient based estimator of MoSo is solely based on intuition. A more rigorous justification would be desirable.
- The initial motivation in the introduction (line 26) states that data pruning ideas can be used to reduce training time. However, within the experimental section, only the accuracy is compared to baseline methods. It would be interesting to see how MoSo compares to baseline methods in terms of training time. Especially because random seems to perform so good and takes essentially no time to score and sample.
- It is mentioned that MoSo is aware of training dynamics (line 57). However, it is not entirely clear what is meant by *awareness* and what part of the method is responsible for this. Furthermore, it is not entirely clear how this is different from other methods that use gradients to prune data and why it should be utilized.
- The conclusion of Proposition 1.2 is not discussed in the main text. It would be interesting to see how close the estimator is to the true criterion. This could even be done in an experimental setting comparable to the ablation in Figure 2(b).

Minor points:
- While Tables 1,2,3 are detailed, the same results are already summarized in Figure 1. Hence, those tables seem redundant if the error bars are included in Figure 1 and the tables could be moved to the supplementary material.
- Initially, it is unclear that MoSo requires to train a surrogate model to estimate the criterion. This should be highlighted earlier and compared to baseline methods.
- Within the experimental section, (all tables and Figure 2(b)) it is unclear how many runs the results are averaged over and if the errors are standard deviations or some other measure of uncertainty.
- Within the experimental section, it is unclear how the hyperparameters were chosen for the baseline methods.
- Algorithm 1, line 1: $\phi$ is not defined/discussed.
- The statement "this suggests that time step sampling is a useful technique for improving the efficiency of our method" in line 305 is not supported by the results in Figure 2(b).

After the Rebuttal:
I have read all other reviews and all rebuttals. Furthermore, I thank the authors for their answers. I keep my original score.

**Questions:**

Questions:
- Q1: In Algorithm 1, line 1: What is $\phi$?
- Q2: How many runs were conducted? Are the aggregated statistics averages over multiple runs? What do the error bars represent? Can you clarify?
- Q3: What is the implication of Proposition 1.2? How close is the gap of the estimator?
- Q4: Regarding the paragraph about robustness to label noise. It is claimed that MoSo "select informative and clean samples" (line 288). However, it is unclear how MoSo is able to distinguish between informative and clean samples as only test accuracies are compared in Table 5. Is it possible to analyze which fraction of the pruned samples are clean and which are informative?
- Q5: Does MoSo suggest an optimal pruning ratio, e.g., by a meaningful cut-off in the distribution of scores?
- Q6: Is it possible to analyze whether MoSo actually selects "representative" samples? For example by comparing how a network trained on the pruned data performs on the full test set.
- Q7: In Figure 2(a), there is a method called "random". It is unclear what this method does. I assume random data points get pruned, i.e., the score is uniform among all data points and so is the selection distribution. Can you clarify?
- Q8: The exact scenario in which training ImageNet would take 45 years (line 152) in unclear. Can you clarify?


Preferential comments:
- When listing many references as in the introduction it is easier to read if they are sorted [1,2,3,4] instead of [4,1,3,2].
- In the results tables bold the best numbers, as well as those with standard errors that overlap with the best method for fair highlighting of the best results.
- Inconsistent presentation of tables. While in Tables 1-3 there are vertical lines between methods, results and avg rank, those lines are missing in Tables 4 and 5.
- The statement "famous influence function" (line 188) may be inappropriate.

Typos/grammar/other:
- Potentially missing related work:
	- Mindermann, S., Brauner, J. M., Razzak, M. T., Sharma, M., Kirsch, A., Xu, W., ... & Gal, Y. (2022, June). Prioritized training on points that are learnable, worth learning, and not yet learnt. In International Conference on Machine Learning (pp. 15630-15649). PMLR.
- References [33] and [34] are the same paper.
- Inconsistent spelling: "core set", "core-set", "coreset"
- Why are there full stops after enumeration marks, i.e., (i). and (ii).?
- Whenever I read "our MoSo" I have the feeling a word is missing, like "our MoSo score" or "our MoSo approach" especially if I expand MoSo (what it stand for).
- lines 117-121, 136, 187: I am not sure why *will* appears here. This is not the case later on, for example in lines 211-215.
- line 123: we focus on **a** classification task?
- line 124: i.i.d. (last dot is missing)
- line 126: I would appreciate if you could specify the spaces of newly introduced variables, i.e., $\delta \in (0,1)$. Same holds for $\eta$.
- line 126: Please introduce $\mathbf{w}$.
- Equations (2) and (6): Is there an empty line in LaTeX before the equation? That would explain the spacing above the equation.
- line 137: Please state once that by $z$ you mean a pair $(x,y)$.
- line 157: Why "could" and not "can"?
- Equation (4) and more: I think the transpose symbol and $T$ are too close. Especially in the supplement where $T$ also appears as an superscript. I would use another transpose symbol, i.e., `\newcommand*{\transpose}{{\mkern-1.5mu\mathsf{T}}}` (looks like $\newcommand*{\transpose}{{\mkern-1.5mu\mathsf{T}}}$) which follows the (DIN) EN ISO 80000-2:2013 standard
- line 187: Why are 1.1 and 1.2 not clickable references?
- line 193: There is no Proposition 1.
- line 203: Please name $I$ as the number of subsets.
- line 252: additional blank space
- line 256: remove 'when'
- line 257: doesn't -> does not
- Sometimes I see $z:(x,y)$ or $\epsilon:0$ in which I read and interpret the colon symbol as an equal sign and I think this notation is confusing for some people.
- It would have helped to start the counters (line numbers, equations, etc) in the supplementary material from the counters of the main paper.
- Appendix Equation (5): $\mathcal{L}$ is a function, not a set (below the sum symbol). I guess $\mathcal{S}$ is meant here. Also, consider putting brackets around the terms to clarify that both terms are within the sum! As mentioned earlier, I consider the notation $\epsilon:0$ confusing.

**Limitations:**

Limitations are only mentioned in the supplementary material but could be more rigorously discussed in the main text.

---

> ### Author Rebuttal · Authors · 2023-08-08
>
> Sincerely thanks for your appreciation of our work. Hoping our response will address your concerns.
>
> ---
>
> **Q1: A rigorous justification of MoSo is desirable.**
>
> A1: Thanks! Proposition 1.2 gives rigorous proof of the approximation accuracy, and we have also updated it in the support material to give a tighter bound. Here, we restate a conclusion:
> $$|M(z) - \hat{M}(z)| \leq  O((\ell \eta + 1)gT + \eta g^2T).$$ MoSo assigns high scores to samples that can reduce the empirical risk. Please refer to Line 112 in the supplementary material for details.
>
> **Q2: How data pruning ideas can be used to reduce training time.**
>
> A2: Thanks! We compare the training efficiency of our MoSo approach with baselines on CIFAR-100 using ResNet50, with 8 Tesla V100 GPUs. MoSo achieves the best trade-off between computational requirements and performance, making it the best-performing model with reasonable computational demands. The overall time cost of MoSo (102.7 min) and re-training the network on the selected subset (61.7 min) is less than directly training a network on the full set (192.2 min). What's more, our method offers more possibilities for the efficient storage of data and the subsequent efficient training of more models [A].
>
> |  Method   | Time-cost (surrogate training)  | Time-cost (scoring)| Time-cost (total)| Accuracy (Pruning-ratio 60%)|
> |  ----  | ----  | ----  | ----  |  ----  |
> | Random  | 0 | 0 |0  | 64.32 |
> | GraNd  | 192.2 m | 683.8 m | 876.0 m  | 60.52 |
> | OPT  | 192.2 m | $\geq 1$day | $\geq 1$day | 58.93 |
> | Moderate  | 192.2 m | **6.63 m** | 198.8 m| 64.92 |
> | MoSo  | **46.5 m** | 56.2 m | 102.7 m | **68.97** (within 61.7 min) |
>
>
> **Q3: What is meant by awareness and what part is responsible for this?**
>
> A3: Thanks! The awareness of training dynamics is to consider the network information at different stages of the training process. The average across different epochs in Proposition 1.1 is responsible for this. Please refer to Figure 2(a) in the paper for the ablation of this part.
>
> **Q4: The difference from other gradient-based methods.**
>
> GraNd is the best-known scheme that also uses gradient information for data pruning. GraNd retains samples with large gradient norms, while our MoSo approach uses the gradient vector which retains more potentially useful information than the norm and approximates the empirical risk.
>
>
> **Q5-1: The conclusion of Proposition 1.2 is not discussed.**
>
> Please refer to A1. And we will add further discussion about this.
>
> **Q5-2: How close the estimator is to the true criterion?**
>
> A5-2: Good question! Because of the word limit, please check Q1 in Global-Rebuttal (https://openreview.net/forum?id=vO6ZdPWaHc&noteId=DsrmXiWeKr) for details.
>
>
> **Q6&7: The same results in Tables 1/2/3 are already summarized in Figure 1. It should be highlighted that MoSo requires training a surrogate model to estimate the criterion.**
>
> A6&7: Thanks, we accept your suggestions and will make modifications accordingly!
>
> **Q8: About the number of runs and the kind of errors.**
>
> A8: The results are the average of 5 independent runs. The standard deviation is selected as the measure of uncertainty.
>
> **Q9: How the hyperparameters were chosen for the baseline methods?**
>
> Thanks! Regarding the experimental settings for the baselines and our MoSo approach, we largely followed the framework used in the Moderate paper for fairness of comparison.
>
> **Q10: That is the meaning of $\phi$.**
>
> A10: $\phi$ means the null set.
>
> **Q11: The statement in line 305 is not supported by Figure 2(b).**
>
> A11: Thanks for the careful review! We will add the computational cost comparison and delete this statement.
>
> **Q12: What is the implication of Proposition 1.2?**
>
> Thanks! Please refer to A1.
>
> **Q13: Is it possible to analyze which fraction of the pruned samples are clean and which are informative?**
>
> A13:  Thanks! This is a good question! We present detailed statistics on TinyImageNet with 20% label noise, totaling 100,000 data samples. After pruning 80% of the data with either MoSo or random selection, we observe MoSo reduces the noise ratio of the retained data (from 20% to 14%).
>
> |  MoSo | Noisy  data| Clean data | Noise Ratio|
> |  ----  | ----  | ----  |  ----  |
> | Retained subset  | 2795  |  17205   |  14% |
> | Discarded (Pruned)   | 17205   |  62795  | 22% |
>
> **Q14: Does MoSo suggest an optimal pruning ratio?**
>
> Ans: Good question! We think it should be artificially defined since it balances efficiency and performance. What MoSo now can do is provide a good pruning suggestion given the pre-defined pruning ratio.
>
>
> **Q15: Whether MoSo actually selects "representative" samples?**
>
> A15: Good question! Here, we compare networks trained on the subset with top-20% largest MoSo scores and the subset with 20% smallest MoSo scores. The former is significantly better than the latter, with a 7% higher top-1 accuracy. This confirms that MoSo-score can reflect the importance of samples to some extent.
>
> |  Subset   | top-1 acc on CIFAR-100|
> |  ----  | ----  |
> | largest 20%| 54.38 |
> | smallest 20%| 47.99|
>
>
> **Q16: In Figure 2(a), what is the method called "random"?**
>
> A16: Thanks! Random (selection) means a completely random selection from a data set. It is a strong baseline as many methods cannot beat it.
>
> **Q17: The exact scenario in which training ImageNet would take 45 years (line 152) is unclear.**
>
> A17: Thanks! Considering the fastest case, before scoring a sample, we choose a small surrogate network with unsupervised initialization and only finetune the last layer. Such a process needs  $>23$ mins. So, the overall time cost for scoring all (1 million) samples is at least about $45$ years.
>
> [A] Cody Coleman, et.al.: Selection via Proxy: Efficient Data Selection for Deep Learning. ICLR-2020.

---

> > ### Comment · Reviewer_DWe5 · 2023-08-15
> >
> > I thank the authors for their answers. They definitely increased the clarity for me.
> >
> > Minor remark:
> > > A10: $\phi$ means the null set.
> >
> > I think $\phi$ (phi) is misleading and there are better symbols for that, e.g., `\emptyset` yields $\emptyset$.

---

> > > ### Author Response · Authors · 2023-08-16
> > > **Sincerely thanks for the response from the Reviewer DWe5**
> > >
> > > We are profoundly grateful for your exceptionally detailed and thoughtful feedback. Your comments are truly invaluable for strengthening the quality of the manuscript. We sincerely appreciate your selfless contributions to the academic community. The professionalism and rigor you demonstrate as a reviewer commands our deepest respect. We promise to thoroughly update the paper according to your suggestions.

---

### Official Review · Reviewer_XiFz · 2023-07-02

**Soundness:** 1 poor
**Presentation:** 2 fair
**Contribution:** 2 fair
**Rating:** 3
**Confidence:** 3

**Summary:**

This paper proposes a framework for data pruning that retains important samples while considering the training dynamics. While the overall methodology relies on analyzing the change in empirical risk from removing individual points, the paper introduces a first-order approximation algorithm that can be efficiently computed. Numerical results demonstrate the effectiveness of the method.

**Strengths:**

-	The motivating idea is intuitive.
-	Numerical results on CIFAR100 and TinyImagenet suggest the method is effective.


**Weaknesses:**

-	The proof for Proposition 1.1 seems to be incorrect. Specifically $L(S/z, w)$ is defined as $L(S,w) – l(z, w) / N$, however this ignores that the empirical risk when we remove a point also needs to be re-normalized to $1/(N-1)$. I believe this breaks the proof steps from page 4 onwards. I suspect that this mis-specification carries on to the remaining theoretical analysis as well.
-	The mathematical presentation is also generally unclear.
 - For example, in the Proof to Proposition 1.1., $L^t$ is used without definition.
 - In Proposition 1.2, the loss function is assumed Lipschitz in parameters for fixed data, but what the loss function is Lipschitz in is not clear.
-	Minor comment: there are also general presentation issues and errors in the main paper, although the authors have caught revisions in the Appendix.


**Questions:**

Please see comments about Proposition 1.1

**Limitations:**

The paper includes a discussion on limitations in the Appendix.

---

> ### Author Rebuttal · Authors · 2023-08-10
>
> Dear reviewer XiFz:
>
> Thank you also for your constructive suggestions. We will carefully address each of your concerns and revise the manuscript accordingly.  If you have any new feedback please do not hesitate to let us know! We will do our best to answer your feedback!
>
> ---
>
> **Q1: The proof for Proposition 1.1 seems to be incorrect. Specifically, $L(S/z, w)$ is defined as L(S, w) - l(z, w)/N, however, this ignores that the empirical risk when we remove a point also needs to be re-normalized to 1/(N-1).**
>
> A1: Thanks! Let me address your concern. In fact, the approximation of this coefficient is not wrong but is common and necessary. We can explain this in ways.
>
>
> 1. Since N is very large in practice (e.g., 1M for ImageNet-1K), hence, the difference between $\frac{1}{N}$ and $\frac{1}{N-1}$ is negligible.
>
> 2. Such an approximation is widely used by many previous works, most notably the ICML-2017 Best Paper [A] and its countless follow-ups. Please check Section 2.1 of the paper [A].
>
> 3. Following your suggestion, we also present an approximator based on the exact coefficient $\frac{1}{N-1}$ according to your suggestion, that is, $$M(z) \approx E_{t}\Big( \frac{\eta_t (2N-3)}{(N-1)^2} ||G^t||^2 - \frac{\eta_t}{(N-1)^2} ||g^t||^2 + \frac{\eta_t(2N-4)}{(N-1)^2} (G^t)^\mathrm{T}g^t\Big),$$ where $G^t = \nabla L(S, w^{t}_S)$, $g^t = \nabla l(z, w^t_S)$, $\eta_t$ is the learning rate, and N is the number of all training data.
>
> We applied the new estimator to data pruning experiments on CIFAR-100, comparing it to the original MoSo estimator. The new estimator behaves similarly to the original one.
>
> |  Estimator   | Pruning-ratio 20%  | Pruning-ratio 40% | Pruning-ratio 60% | Pruning-ratio 80%|
> |  ----  | ----  | ----  | ----  |  ----  |
> | Original estimator in the paper | 75.76 | **74.29** | **68.97**  | 54.38 |
> | New estimator derived here | **75.81** | 73.95 | 68.48  | **54.45** |
>
>
>
> **Q2-1: The unclear mathematical presentation: in the Proof to Proposition 1.1., $L^t$ is used without definition.**
>
> A2-1: Thank you for the feedback. The loss functions $L(\cdot)$ and $l(\cdot)$are defined in Line 158 of the main paper. The superscript t refers to the t-th training epoch, as noted in Line 94 of the supplementary material. $L^t$ represents the loss value at epoch t. We will clarify these definitions in the revised paper.
>
>
> **Q2-2: The unclear mathematical presentation: In Proposition 1.2, the loss function is assumed Lipschitz in parameters for fixed data, but what the loss function is Lipschitz is not clear.**
>
> A2-2: Thank you for pointing this out. The proof relies only on the Lipschitz-continuity assumption for parameters, not the specific loss function. This suggests MoSo could apply more broadly, like data reduction for multimodal pretraining. In the paper, we state the loss function used is cross-entropy (Lines 125/133/138 in the main paper). The proof's generality indicates MoSo's potential beyond classification tasks. We will clarify that the proof holds for any continuously differentiable loss function satisfying Lipschitz continuity.
>
> **Q3: Minor comment: there are also general presentation issues and errors in the main paper, although the authors have caught revisions in the Appendix.**
>
> A3: Thanks. We will further check and polish the presentation in the revised paper.
>
>
> [A] Koh P W, Liang P. Understanding black-box predictions via influence functions. ICML-2017.

---

> > ### Comment · Reviewer_XiFz · 2023-08-14
> > **Thanks for the response**
> >
> > Thanks for the response. My concern with the $1/N \approx 1/(N-1)$ approximation is that in the original influence functions paper [A], this is not used for a rigorous proof (c.f. Proposition 1.1). However in this draft, M(z) given in definition 1 is different from M(z) redefined in line 94-95 of the Supplement. I agree that this difference may be small in practice and I commend the authors for deriving a corrected version of the approximation and for running some experiments showing that the corrected version does not yield a major difference from the initial. I encourage the authors to revise the paper, for example by beginning with the rigorous version and then demonstrating approximations or by forgoing the theoretical statements and presenting the approximator via mathematical steps.

---

> > > ### Author Response · Authors · 2023-08-19
> > > **Thanks for the comments**
> > >
> > > We sincerely appreciate your time and efforts in reviewing our work, which helps improve our paper. Regarding your concerns, we would like to make the following further clarifications. We hope our response addresses your concerns and look forward to discussing them with you.
> > >
> > > ---
> > >
> > > **Q1. The coefficient approximation is used in the original influence functions paper [A], however, it is not used for rigorous proof.**
> > >
> > > The original influential work uses this approximation in deriving influence functions but does not provide a theoretical analysis of its tightness. However, subsequent papers have rigorously employed it in formal proofs:
> > >
> > > Paper [B] provides error bounds on influence estimate accuracy leveraging this approximation.
> > >
> > > Paper [C] theoretically shows influence functions can identify samples to relabel for lower test risk, relying on this simplified coefficient.
> > >
> > > Work [D] formally relates worst-case risk change rates to single sample loss perturbation using this approximation.
> > >
> > > And OPT [E] derives generalization error bounds for influence-based data pruning, founded on this established approximation.
> > >
> > > In summary, while tightness was not analyzed initially, many papers since have formally adopted this approximation within rigorous mathematical proofs and bounds. This demonstrates its acceptance in theoretical analyses, beyond just the initial empirical derivation. Our work aligns with this trend of employing the approximation in formal contexts, rather than solely heuristically.
> > >
> > >
> > > ---
> > >
> > > **Q2. I encourage the authors to revise the paper, for example by beginning with the rigorous version and then demonstrating approximations or by forgoing the theoretical statements and presenting the approximator via mathematical steps.**
> > >
> > > We appreciate your thoughtful suggestions! As a compromise, we will revise the paper to include the coefficient approximation in the tightness proof for Proposition 1.2, while still ensuring overall simplicity. This yields the following new tightness bound:
> > >
> > > $$O( |\mathcal{M}(z) - \hat{\mathcal{M}}(z)|) \leq {O}\Big( (\ell\eta + 1) g T + \eta g^2 T (1 + \frac{3}{N})    \Big). $$
> > >
> > > Compared to the original Proposition 1.2 bound, this just introduces one additional negligible term $\frac{3}{N} \eta g^2 T$, since $N$ is typically very large.
> > >
> > > By incorporating the approximation only in the key tightness analysis, we aim to strike a balance between mathematical rigor and manuscript clarity/conciseness. The impact on the final bound is marginal, yet it formally addresses the coefficient concern. Please let us know if you feel this targeted revision satisfactorily resolves the approximation issue while maintaining readability. We appreciate you working with us to improve the paper while preserving its accessible style.
> > >
> > >
> > > ---
> > >
> > > [A] Koh P W, Liang P. Understanding black-box predictions via influence functions. ICML-2017.
> > >
> > > [B] Zhifeng Kong, Kamalika Chaudhuri. Understanding Instance-based Interpretability of Variational Auto-Encoders. NeurIPS-2021.
> > >
> > > [C] Shuming Kong, Yanyan Shen, Linpeng Huang. Resolving Training Biases via Influence-based Data Relabeling. ICLR-2022
> > >
> > > [D] Zifeng Wang, et. al. Less Is Better: Unweighted Data Subsampling via Influence Function. AAAI-2022
> > >
> > > [E] Shuo Yang, et. al. Dataset Pruning: Reducing Training Data by Examining Generalization Influence. ICLR-2023

---

> > > ### Author Response · Authors · 2023-08-21
> > > **Looking forward to your further reply**
> > >
> > > Dear Reviewer XiFz:
> > >
> > > We sincerely thank you for your efforts in reviewing our paper and your suggestions for polishing the manuscript. As we are approaching the end of the discussion period, we would like to ask whether there are any remaining concerns regarding our paper or our response. We are happy to answer any further questions.
> > >
> > > Best regards,
> > > Submission909 Authors

---

### Official Review · Reviewer_QDiJ · 2023-07-02

**Soundness:** 3 good
**Presentation:** 3 good
**Contribution:** 3 good
**Rating:** 6
**Confidence:** 4

**Summary:**

This paper presents MoSo, a method to identify and remove the least informative samples from a large dataset. The underlying idea is to consider the impact of each sample on the optimal empirical risk. Quantifying this exactly requires leave-one-out retraining for every point, which is intractable. So, the authors provide an approximation of this score that is more efficient in that it does not require leave-one-out retraining for every point. They provide bounds on the quality of this approximation and present empirical comparisons of the data pruning approach with contemporary baselines.

**Strengths:**

* The proposed approximation of the MoSo score and the accompanying analysis are sound and novel to the best of my knowledge.
* The authors present theoretical results establishing the accuracy of their approximation.
* Empirical evaluations on benchmark vision tasks are provided that support the effectiveness of the method relative to state-of-the-art baselines.

**Weaknesses:**

* The authors state that data pruning can address the computational challenges in the introduction. However, it is not clear to me how the method can provide a computational speedup given that it needs to train a surrogate model on the entirety of the dataset (Line 8 of Algorithm 1) to compute the MoSo score approximation.
* The claimed asymptotic computation times are difficult to understand. Upon looking at the algorithm, it seems that the power of the approximation in Eq. (4) comes from the fact that the surrogate network needs to only be trained once, rather than once per each point (as required by Eq. 3). This should be clarified. Please see the Questions section for more details.
* The claim that randomly sampling a few time steps rather than considering all $T$ steps “reduces the overall complexity to be less than $\mathcal O(Tn)$” does not seem sound since the expectation of a uniform sample from $\\{1, \ldots, T\\}$ is $(T+1)/2$.
* Only the ResNet50 architecture is used exclusively throughout the experiments to construct the pruned datasets. Diversity of architecture in the evaluations would have strengthened the method’s appeal.
* The computational complexity of the method is quite high as it requires training the model on the entire dataset relative to the compared methods.


**Questions:**

1. It is not clear to me why the original MoSo score “takes $\mathcal O(Tn^2)$ time.” More generally, the asymptotic complexities are confusing throughout because neither the batch size of SGD, nor the dimensionality of the data points $d$ are accounted for in the asymptotic analysis. Are we assuming that the batch size = $n$, i.e., regular GD and dimensionality = 1 for the samples?
2. Related to the above, I don’t understand how the approximation in Eq. (4) is faster in an asymptotic sense than the original MoSo score in Eq. (3). Assuming that batch size = $n$ and ignoring the dimensionality of the points as the authors do, two full rounds of training to obtain $w_{\mathcal S}^*$ and $w_{\mathcal S \setminus z}^*$ take $\mathcal O(n T)$ time overall (based on the way the authors express training time). Once we have those two models, computing the difference of losses in Eq. (3) takes $\mathcal O(n)$ time. Where is the quadratic in $n$ coming from? It would help to clarify that Eq. (4) requires only training the model once (as in Line 8 of Alg. 1), unlike Eq. (3) which requires computing $w_{\mathcal S \setminus z}^*$ for each $z$.
3. What is the appeal of the data pruning method from a practical efficiency perspective, given that it requires training the surrogate network for $(T+1)/2$ iterations in expectation on the full dataset?


**Limitations:**

Yes, they are mentioned in the supplementary material.

---

> ### Author Rebuttal · Authors · 2023-08-10
>
> Dear reviewer QDiJ:
>
> Thanks for your time and efforts in reviewing our paper. We will address your concerns below.
>
> ---
>
> **Weaknesses**
>
> **Q1: It is not clear to me how the method can provide a computational speedup.**
>
> A1: Thanks for your suggestion. We evaluated MoSo and the other baseline methods on a server with 8 Tesla V100 GPUs. We used the CIFAR-100 dataset and the ResNet50 backbone for our experiments.
>
> MoSo achieves the best trade-off between computational requirements and performance, making it the best-performing model with reasonable computational demands. Notably, it outperforms the state-of-the-art method, Moderate, while being more efficient.
>
> The overall time cost of MoSo (102.7 min) and re-training the network on the selected subset (61.7 min) is less than directly training a network on the full set (192.2 min). What's more, our method offers more possibilities for the efficient storage of data and the subsequent efficient training of more models.
>
>
>
> |  Method   | Time-cost (surrogate training)  | Time-cost (scoring)| Time-cost (total)| Accuracy (Pruning-ratio 60%)|
> |  ----  | ----  | ----  | ----  |  ----  |
> | Random  | 0 | 0 |0  | 64.32 |
> | GraNd  | 192.2 m | 683.8 m | 876.0 m  | 60.52 |
> | OPT  | 192.2 m | $\geq 1$day | $\geq 1$day | 58.93 |
> | Moderate  | 192.2 m | **6.63 m** | 198.8 m| 64.92 |
> | MoSo  | **46.5 m** | 56.2 m | 102.7 m | **68.97** (within 61.7 min) |
>
>
>
> **Q2: The claimed asymptotic computation times are difficult to understand. It is not clear why the original MoSo score “takes $O(Tn^2)$ time and how the approximation in Eq. (4) is faster.**
>
> A2: We will clarify this below. Hope it can address your concern. MoSo without approximation requires $n$ times full training of the network, with each training a network on $n-1$ data samples for $T$ epochs. Consequently, it has a theoretical complexity of $O(Tn(n-1)) \approx O(Tn^2)$. Our MoSo estimator avoids this costly leave-one-out retraining. It only require training a network on the full dataset ($n$ data samples) for $T$ epochs. Thus, the time complexity is substantially reduced to $O(Tn)$, providing a notable gain in efficiency.
>
>
> **Q3: The claim that randomly sampling a few time steps rather than considering all T steps “reduces the overall complexity to be less than O(Tn)” does not seem sound since the expectation of a uniform sample from {1,...,T} is (1+T)/2.**
>
> A3: We would like to clarify that the time complexity primarily concerns the number of iterations needed to compute the required gradients in Equation 4, which is the most time-consuming process. Assuming the surrogate network is trained with T steps (T=50), our MoSo estimator requires estimating the gradients at each iteration, resulting in an O(Tn) time cost. In our implementation, we found that this process can be accelerated by randomly sampling t steps (t=10,t<T) from all the T steps (Line 231) to perform gradient estimation and take their average. So the actual complexity is O(tn) which is less than O(Tn) as t<T. Figure 2b in the paper shows the effect of the sampling ratio (t/T), demonstrating that such a sampling operation trades a slight drop in performance for a 5-fold increase in speed.
>
>
> **Q4: Only the ResNet50 architecture is used exclusively throughout the experiments to construct the pruned datasets. Diversity of architecture in the evaluations would have strengthened the method’s appeal.**
>
> A4: Thanks! According to your suggestion, we conduct experiments on various architectures and report the results in the following table. Notably, the generalization ability of MoSo is well since the data is scored by using ResNet-50 as the surrogate network.
>
>
> |  Network | Dataset  | Random (PR: 80%)| MoSo (PR: 80%) |
> |  ----  | ----  | ----  |  ----  |
> | GoogleNet  | C-100 | 59.2 |  62.37 |
> | MobileNetV2  | C-100 | 49.25 |  51.31 |
> | DenseNet121  | C-100 | 56.92 |  57.49 |
> | Swin-T  | IN-1K | 67.20 |  72.66 |
>
>
>
> **Q5: The computational complexity of the method is quite high as it requires training the model on the entire dataset relative to the compared methods.**
>
> A5: Pre-training a surrogate network before pruning is a commonly-used procedure. Almost all the compared methods except random selection require training a surrogate network on the full dataset before pruning. The previous pruning methods need to train the surrogate network until convergence, normally using the same number of epochs as training on the pruned dataset (e.g., 200 epochs on CIFAR-100). Notably, our MoSo does not require full convergence of the surrogate network, for example, we only use 50 epochs on CIFAR100 instead of the original 200 epochs. In summary, our MoSo achieves better performance with less computational cost.
>
>
> **[Questions]. Why the original MoSo score “takes $O(Tn^2)$ time and how the approximation in Eq. (4) is faster?**
>
> Thank you for the feedback. Please refer to A2.
>
>
> **[Q6] What is the appeal of the data pruning method from a practical efficiency perspective**
>
> Thanks for your comments! First, I think there must be some misunderstanding about “MoSo need (T+1)/2 training epochs”. Please refer to A3, which will address your question.
>
> With the smaller but informative subset selected by MoSo (e.g. only 20% of the data), we can do many things, like reducing storage overhead, decreasing training cost for training subsequent models [A], and even enabling continual learning [B].
>
> [A] Cody Coleman, et.al.: Selection via Proxy: Efficient Data Selection for Deep Learning. ICLR-2020.
>
> [B] Jachong Yoon, et.al.: Online Coreset Selection for Rehearsal-based Continual Learning. ICLR-2022.

---

> > ### Comment · Reviewer_QDiJ · 2023-08-10
> >
> > Thank you for your response. I read the other reviews, your responses to them, and the general response. In light of the clarifications and compelling experimental results regarding computational efficiency of the method, I decided to raise my score to a 6.

---

> > > ### Author Response · Authors · 2023-08-11
> > > **A Grateful Response to Reviewer QDiJ**
> > >
> > > We are truly grateful for your encouraging feedback. It means a great deal to know our work resonated positively. We welcome any additional questions you may have during the discussion period and are more than happy to provide clarification. Furthermore, we will continue refining the evaluation section to more clearly convey our contributions. Our aim is to produce the highest quality work that lives up to the standards of yourself and the broader research community. Thank you again for recognizing our efforts - it inspires us to keep improving. We sincerely appreciate you taking the time to provide such thoughtful and constructive feedback.

---

### Official Review · Reviewer_eYrR · 2023-07-06

**Soundness:** 3 good
**Presentation:** 3 good
**Contribution:** 3 good
**Rating:** 7
**Confidence:** 3

**Summary:**

This paper presents moving-one-sample-out (MoSo) for the purpose of coreset selection. This algorithm measures how empirical loss changes when excluding individual points during training. The paper introduces an approximation and other tricks to make this method computationally feasible. Experimentally, MoSo outperforms other methods on standard datasets, and also has nice properties like generalization to other architectures and robustness to label noise.

**Strengths:**

- Good presentation: method is clear and contextualized properly through related work and comparison experiments
- Good results: MoSo does better on all datasets evaluated than the comparison baselines
- Method seems new and is differentiated from related work
- Interesting ablations on architecture generalization and label noise

**Weaknesses:**

- Lack of analysis on the computational cost of this method. Comparisons use same amount of samples seen during training, but this does not take into account the extra cost of calculating the coreset for some of these methods. This can be addressed with end-to-end training time or a similar metric. Though the authors mention that this is substantially cheaper than methods that require training a full-network for scoring, this information is still important, especially when considering scaling up this method to larger datasets.

**Questions:**

See weaknesses

Is there a reason why experiments aren't done on CIFAR-10? This seems to be a standard benchmark for this line of work.

Would also be interested in seeing this method evaluated on a larger dataset or other tasks (e.g. large-scale multi-modal learning mentioned in supplementary)

**Limitations:**

Addressed in weaknesses - lack of analysis on computational cost.

---

> ### Author Rebuttal · Authors · 2023-08-10
>
> Dear reviewer eYrR:
>
>
> Thank you for appreciating our approach. We will address your comments below.
>
> ----
>
> **Q1: Lack of analysis of the computational cost of this method.**
>
> A1: This is a good question!  We will incorporate this experiment into the paper. We evaluated MoSo and the other baseline methods on a server with 8 Tesla V100 GPUs. We used the CIFAR-100 dataset and the ResNet50 backbone for our experiments.
>
> MoSo achieves the best trade-off between computational requirements and performance, making it the best-performing model with reasonable computational demands. Notably, it outperforms the state-of-the-art method, Moderate, while being more efficient. Because of the use of large-scale linear programming in the scoring phase, OPT is significantly more time-consuming than the other methods.
>
> |  Method   | Time-cost (surrogate training)  | Time-cost (scoring)| Time-cost (total)| Accuracy (Pruning-ratio 60%)|
> |  ----  | ----  | ----  | ----  |  ----  |
> | Random  | 0 | 0 |0  | 64.32 |
> | GraNd  | 192.2 m | 683.8 m | 876.0 m  | 60.52 |
> | OPT  | 192.2 m | $\geq 1$day | $\geq 1$day | 58.93 |
> | Moderate  | 192.2 m | **6.63 m** | 198.8 m| 64.92 |
> | MoSo  | **46.5 m** | 56.2 m | 102.7 m | **68.97** |
>
> ---
>
> **Questions.**
>
> **Q2: Is there a reason why experiments aren't done on CIFAR-10? This seems to be a standard benchmark for this line of work.**
>
> A2: We primarily followed the experimental setup in the recent method, Moderate [A]. Here, we have conducted experiments on CIFAR-10 and present the results in the Table below. Our method demonstrates comparable performance to existing methods at low pruning ratios and surpasses them at high pruning ratios.
>
> It's worth noting that CIFAR-10 is a smaller dataset compared to CIFAR-100, Tiny-ImageNet, and ImageNet, which are evaluated in our paper. Our method exhibits superior performance on larger datasets, effectively showcasing the potency of our model. In the case of small-scale datasets like CIFAR-10, there isn't a compelling reason to employ pruning.
>
>
> |  Method   | 20%  | 40% |  60% | 80% |
> |  ----  | ----  | ----  | ----  |  ----  |
> | Random  | 93.05 | 92.19 | 89.77  | 85.20 |
> | Forgetting  | 94.52 | 93.33 | 91.41  | 86.12 |
> | EL2N  | **94.59** | 93.77 | 92.24 | 85.23 |
> | Moderate  | 94.05 | **93.81** | **93.10** | 86.05   |
> | MoSo  | 94.20 | 93.60 | 93.05 | **86.26** |
>
>
> **Q3: Would also be interested in seeing this method evaluated on a larger dataset or other tasks (e.g. large-scale multi-modal learning mentioned in supplementary).**
>
>
> A3: We have conducted experiments on the CC3M dataset, which contains 3 million image-text pairs, to train a CLIP [B] model using two backbone architectures: ResNet50 and ViT-B.
>
> Following CLIP [B], we evaluate the trained model in zero-shot image classification. The result shown in the Table below demonstrates notable improvements over the random selection baseline. Notably, after removing 80% of the data, we observe a 3.2% increase in performance. Furthermore, data selected using the ResNet50 backbone also enhances the performance of the transformer-based architecture, ViT-B, outperforming the random baseline by 2.6%. This showcases the generalization ability of the pruned data.
>
>
> |  Method   | Training Data | Zero-shot classification on ImageNet |
> |  ----  | ----  | ----  |
> | CLIP (R50) [B] | Full dataset | 16.7  |
> | CLIP (R50) | Random selection 20% subset  | 5.9 |
> | MoSo (R50) | MoSo (ours) Selection 20% subset using CLIP(R50) | 9.1(+3.2) |
> | CLIP (ViT-B) [B] | Full dataset | 16.1  |
> | CLIP (ViT-B) | Random selection 20% subset  | 5.5 |
> | CLIP (ViT-B) | MoSo (ours) Selection 20% subset using CLIP(R50)  | 8.1(+2.6) |
>
>
> [A]. Xiaobao Xia, et.al.: Moderate Coreset: A Universal Method of Data Selection for Real-world Data-efficient Deep Learning. ICLR-2023
> [B]. Alec Radford, et.al.: Learning Transferable Visual Models From Natural Language Supervision. ICML-2021

---

> > ### Comment · Reviewer_eYrR · 2023-08-19
> >
> > Thank you for your thorough response. I have adjusted my score as my concerns have been addressed.

---

> > > ### Author Response · Authors · 2023-08-19
> > > **Sincerely thanks for the response!**
> > >
> > > We sincerely thank the reviewer for their generosity in time and feedback, and we are also incredibly grateful to the reviewer for their willingness to reconsider and increase their score after reviewing our detailed response! We are excited to continue refining the work guided by the reviewer's suggestions!

---

### Official Review · Reviewer_xfpA · 2023-07-06

**Soundness:** 3 good
**Presentation:** 3 good
**Contribution:** 2 fair
**Rating:** 5
**Confidence:** 3

**Summary:**

This paper proposes a data-pruning method. The authors argue that sample importance should not be determined by sample difficulty. Alternatively, they present the MoSo score, which quantifies the changes in empirical risk upon excluding a single data point. An efficient approximation for MoSo is proposed to calculate the score with some theoretical guarantee. Experiments are conducted with common image classification benchmarks.

**Strengths:**

[1] The proposed measure of sample importance is intuitive, and the approximation appears novel.

[2] The paper is overall well-written, with a clear description of the insights and adequate related work.

**Weaknesses:**

[1] The authors propose an approximation for the MoSo score.  However, it is unclear from the experiment how expensive the proposed method is compared to the baselines.



**Questions:**

[1] I appreciate the set of experiments with label noise, as it could provide evidence for the authors' insight on difficult data. I would like to see the overlapping ratio between samples with injected noise and pruned data. This overlapping can be more informative than simple test accuracy in demonstrating whether the proposed method can exclude hard but noisy data.

[2] The authors mainly report the trade-off of top-1 accuracy and data pruning ratio, which could demonstrate the effectiveness of a method. It would be interesting to see the data pruning ratio across classes to better understand the method. Does the pruning change the data sample distribution, and does the method improve/hurt certain classes?

**Limitations:**

The authors didn't discuss limitations in the paper.

---

> ### Author Rebuttal · Authors · 2023-08-10
>
> Dear reviewer xfpA:
>
> Thank you for your comments which help us improve our work!
>
> ----
>
> **Q1: How expensive the proposed method is compared to the baselines?**
>
> A1: Thanks for your suggestion. We will incorporate the additional results into the paper. We evaluated MoSo and the other baseline methods on a server with 8 Tesla V100 GPUs. We used the CIFAR-100 dataset and the ResNet50 backbone for our experiments.
>
> MoSo achieves the best trade-off between computational requirements and performance, making it the best-performing model with reasonable computational demands. Notably, it outperforms the state-of-the-art method, Moderate, while being more efficient. Because of the use of large-scale linear programming in the scoring phase, OPT is significantly more time-consuming than the other methods.
>
> |  Method   | Time-cost (surrogate training)  | Time-cost (scoring)| Time-cost (total)| Accuracy (Pruning-ratio 60%)|
> |  ----  | ----  | ----  | ----  |  ----  |
> | Random  | 0 | 0 |0  | 64.32 |
> | GraNd  | 192.2 m | 683.8 m | 876.0 m  | 60.52 |
> | OPT  | 192.2 m | $\geq 1$day | $\geq 1$day | 58.93 |
> | Moderate  | 192.2 m | **6.63 m** | 198.8 m| 64.92 |
> | MoSo  | **46.5 m** | 56.2 m | 102.7 m | **68.97** |
>
> ----
>
> **Q2: The overlapping ratio between samples with injected noise and pruned data.**
>
> A2:  Thanks! This is a good question! We present detailed statistics on TinyImageNet with 20% label noise, totaling 100,000 data samples. After pruning 80% of the data with either MoSo or random selection, we observe MoSo meaningfully reduces the noise ratio of the retained data (decreasing from 20% to 14%).
>
> |  MoSo | Noisy  data| Clean data | Noise Ratio|
> |  ----  | ----  | ----  |  ----  |
> | Retained subset  | 2795  |  17205   |  14% |
> | Discarded (Pruned)   | 17205   |  62795  | 22% |
>
>
> **Q3: Does the pruning change the data sample distribution, and does the method improve/hurt certain classes?**
>
> A3:  This is a very important question, and we have visualized and analyzed the relevant statistical results, which will be included in the revised paper in the future. See Q2 in Global-Rebuttal (https://openreview.net/forum?id=vO6ZdPWaHc&noteId=DsrmXiWeKr) for details.
>
>
> **Q4: The authors didn't discuss limitations in the paper.**
>
> A4: Please refer to Sec. 5. in the supplementary material, where we discussed the limitations and future works.

---

> > ### Comment · Reviewer_xfpA · 2023-08-14
> > **Rebuttal Response**
> >
> > Additional experiments and visualization provide further details to understand the proposed method. Thank the author's response, and I am happy to increase my score.

---

> > > ### Author Response · Authors · 2023-08-16
> > > **A Grateful Response to Reviewer xfpA**
> > >
> > > We sincerely thank the reviewer for taking the time to thoroughly review our additional experiments and visualizations. The reviewer's openness to increasing their score after considering our response is greatly encouraging. We are motivated to continue improving the work based on this thoughtful feedback.

---

### Author Rebuttal · Authors · 2023-08-10

**Q1: How close the estimator is to the true criterion?**

A1: This is a good question! We will add this to the revised paper!

We compare the approximation error comparison between ours and the well-known influence function, please refer to Figure 1 in the PDF attachment. Our method exhibits better approximation performance, with a Spearman correlation of 0.45041617686 to the real MoSo value, while the influence function is only with a Spearman correlation of 0.03515931916 to the real value.

**Q2: Does the pruning change the data sample distribution, and does the method improve/hurt certain classes?**

A2: Thanks for the constructive question! We will add this to the revised paper!

We visualize the class-wise accuracy before and after applying our MoSo data pruning approach, please refer to Figure 2 in the PDF attachment. We can observe that the correlation between the two is very significant, where the Spearman correlation coefficient is 0.9133613248, and the P value is 0.0295558770. This shows that the performances before and after being pruned with MoSo are quite consistent, and no significant improvement or harm to a particular category was observed.

We investigated whether the data in each category was balanced after applying our MoSo data pruning approach, please refer to Figure 2 in the PDF attachment.
Ideally, in the most balanced case, the number of data in each category is 100.
It is not difficult to find that the number of data categories is quite balanced for our approach, where the mean is 99.99, var is 9.006.

---

### Decision · Program_Chairs · 2023-09-21

**Decision:**

Accept (poster)

**Comment:**

Finding efficient ways to approximate the importance of individual data points is a critical problem. However, it is very expensive to perform methods like memorization score which require performing a variant of leave-one-out training. This paper proposes MoSo, which approximates the importance of individual data points by measuring the correlation between the gradient of each data point and the average gradient. Using this criterion, the authors demonstrated improved performance as data are removed, particularly for high pruning rates.

Reviewers generally found the paper's topic important, experiments well done and results compelling. I recommend acceptance.